nanotechnology

rapid propagation of charge carriers, criss-crossed $TiO_2$ nanoflakes, photoreduction $CO_2$

**Author for correspondence:**
Juangang Wang
e-mail: shanxiwangjuangang@126.com

This article has been edited by the Royal Society of Chemistry, including the commissioning, peer review process and editorial aspects up to the point of acceptance.

# Enhanced photoreduction CO₂ efficiency by criss-crossed TiO₂ nanoflakes combined with CdS under visible light

## Fei Wang, Juangang Wang and Yunhuan Cheng

Anhui Key Laboratory of Energetic Materials, College of Chemistry and Material Science, Huaibei Normal University, Huaibei 235000, Anhui, People's Republic of China

  FW, 0000-0001-7201-2500; JW, 0000-0003-3163-2804

In this paper, a novel photocatalyst CNCC with excellent visible light photocatalytic performance was successfully prepared to optimize the $CO_2$ photoreduction performance. The results showed that the methanol formation rate of CNCC was 24.7 $\mu$mol g$^{-1}$ h$^{-1}$, which was 1.42 times higher than that of NCC. The enhanced photoactivity is attributed to the rapid propagation of charge carriers induced by light from the constructed composite structure.

## 1. Introduction

Photocatalysis is increasingly being seen as a potential alternative to solar fuel production [1–4]. In addition to the formation of hydrogen from water, another focus is photocatalytic reduction in $CO_2$. Because of the technical difficulties associated with hydrogen storage, products [5–8] from $CO_2$, such as methane and methanol, can easily be used as current energy sources. It is well known that one of the most significant challenges in photoelectrochemical processes is highly efficient separation and transmission of photoinduced electron–hole pairs. To suppress a recombination and improve transmission of electron–hole pairs, researchers have designed many schemes. A novel photocatalyst $Ag_3PO_4$@MWCNTs@PANI and Z-scheme heterojunction photocatalyst $Ag_3PO_4$@MWCNTs@Cr:SrTiO$_3$ were successfully prepared by a facile *in situ* precipitation method [9,10]. The MWCNTs penetrating in the bulk phase of $Ag_3PO_4$ could serve as conductors of photogenerated electrons and rapidly migrated electrons to the surface of the photocatalysts. Han *et al*. [11] synthesized uniform spherical CdS/$TiO_2$ core–shell nanoparticles with different $TiO_2$ shell thicknesses, generating an energy gradient at the interface to spatially separate the electrons and holes. Zhan *et al*. [12] have prepared $TiO_2$ nanorod films on FTO substrates, which exhibit a longer electron lifetime and more

royalsocietypublishing.org/journal/rsos　R. Soc. open sci. 6: 181789

effective separation of photogenerated electron–hole pairs. A highly efficient Pt–$TiO_2$ nanostructured film, with fast electron-transfer rate and efficient electron–hole separation by the Pt nanoparticles, is reported [13]. In this paper, we propose a material model of criss-crossed $TiO_2$ nanoflakes combined with CdS (CNCC) and compared its photocatalytic activity with that of the usual $TiO_2$ nanoparticles with CdS (NCC). The geometry structure of criss-crossed $TiO_2$ nanoflakes is conductive to charge separation, therefore inhibiting recombination can significantly improve its $CO_2$ photoreduction performance.

# 2. Experimental methods

All chemicals used in the experiment (Tianjin Chemical Reagent Company) are analytical reagent grade.

The FTO glass ($5 \times 5$ cm$^2$) was cleaned, and then coated with $TiO_2$ nanowire in ethanol solution. FTO glass was placed in a tetrafluoroethylene reactor (200 ml), 110 ml ethanol, 5 ml deionized water, 1 mmol $Ti(OC_2H_5)_4$, 1 mmol trihydroxytriethylamine, 1 mmol urea glycoside and 1 mmol hexadecanol were added, and then placed in an oven at 180°C for 7 days. The substrate was washed and baked at 450°C for 50 min to optimize the photoreduction performance of $CO_2$.

The CdS–$TiO_2$ heterostructure photocatalysts were prepared using a simple precipitation method. The FTO film of prepared interconnected $TiO_2$ nanowires was dispersed in a certain volume of 0.1 M Cd (NO3)$_2$ aqueous solution, using the same capacity of 0.1 M $Na_2S$ aqueous solution as the precipitator introduced by dripping slowly. Then the film was rinsed several times with deionized water. The anatase $TiO_2$ nanoparticles (15 nm primary particle size) were purchased from Aladdin. The synthetic procedures of $TiO_2$ nanoparticles combined with CdS were almost the same as the procedures of criss-crossed $TiO_2$ nanoflakes combined with CdS.

# 3. Results and discussion

## 3.1. Morphological characteristics and phase structures

The morphology of $TiO_2$ on FTO glass was studied by SEM. Figure 1a indicates the micrometre-sized $TiO_2$ film. A magnified SEM image in figure 1b clearly demonstrates the highly organized structure of the film packaging by criss-crossed $TiO_2$ nanoflakes, which provide more surface area and active sites for the next catalytic reaction [14,15]. Figure 1c displays magnified SEM of the criss-crossed $TiO_2$ nanoflakes covered with CdS-nanosized crystallites. The CdS particle size is about 8 nm. Transmission electron microscopy (TEM) images for $TiO_2$ and $TiO_2$–CdS composite are presented in figure 1d,e to explore the nanostructure. As shown in figure 1d, criss-crossed nanoflakes were distinctly observed. From figure 1e, it can be seen that the porous surface is composed of nanoparticles with a diameter of less than 10 nm. EDX mapping images (figure 1f–i) indicate that the sample contains Ti, O, Cd and S; this finding further confirms the coexistence of titanium dioxide and cadmium sulfide. Ti, O, Cd and S are well dispersed in the samples; thus, CdS nanoparticles are uniformly dispersed on the surface of $TiO_2$. Their phase structures were studied by XRD. Figure 1j shows the XRD diagram of criss-crossed $TiO_2$ nanoflakes, indicating formation of the anatase phase. The diffraction peak of $TiO_2$ can be seen by observing the XRD diagram of CdS@$TiO_2$ composite. The peaks located at 26.5°, 44.4° and 52° could be indexed to the (111), (220) and (311) crystal planes of cubic CdS phase, respectively.

## 3.2. Transportation time and recombination time constants

The transmission and recombination of photoinduced electrons are the main determinants of the efficiency of $CO_2$ photoreduction; thus, the study of these effects in CNCC is of great significance for the further development of the $CO_2$ photoreduction process. Intensity-modulated photocurrent spectroscopy was used to measure the transmission characteristics and intensity-modulated photovoltage spectroscopy was used to measure recombination characteristics [16]. Figure 2a compares the transmission time constants of $TiO_2$ nanoflakes and nano-titanium dioxide particles combined with CdS as light intensity functions. The transmission time constant $\tau_c$ of CNCC is $2.24 \times 10^{-4}$ s at the light intensity ($9.12 \times 10^{16}$ cm$^{-2}$ s$^{-1}$) and $3.31 \times 10^{-3}$ s at the light intensity ($1.15 \times 10^{15}$ cm$^{-2}$ s$^{-1}$). Meanwhile, the transmission time constant $\tau_c$ of NCC is $8.34 \times 10^{-4}$ s at $9.19 \times 10^{16}$ cm$^2$s$^{-1}$ and $8.04 \times 10^{-3}$ s at $1.18 \times 10^{15}$ cm$^{-2}$ s$^{-1}$. The electron transmission between $TiO_2$ nanoparticles and CdS is slower than that of CNCC film; this may be due to the electron's residence time in the trap of the particle network (for example, the number of connections between particles) and the region of contact between particles that limits it [17]. That is to say, the

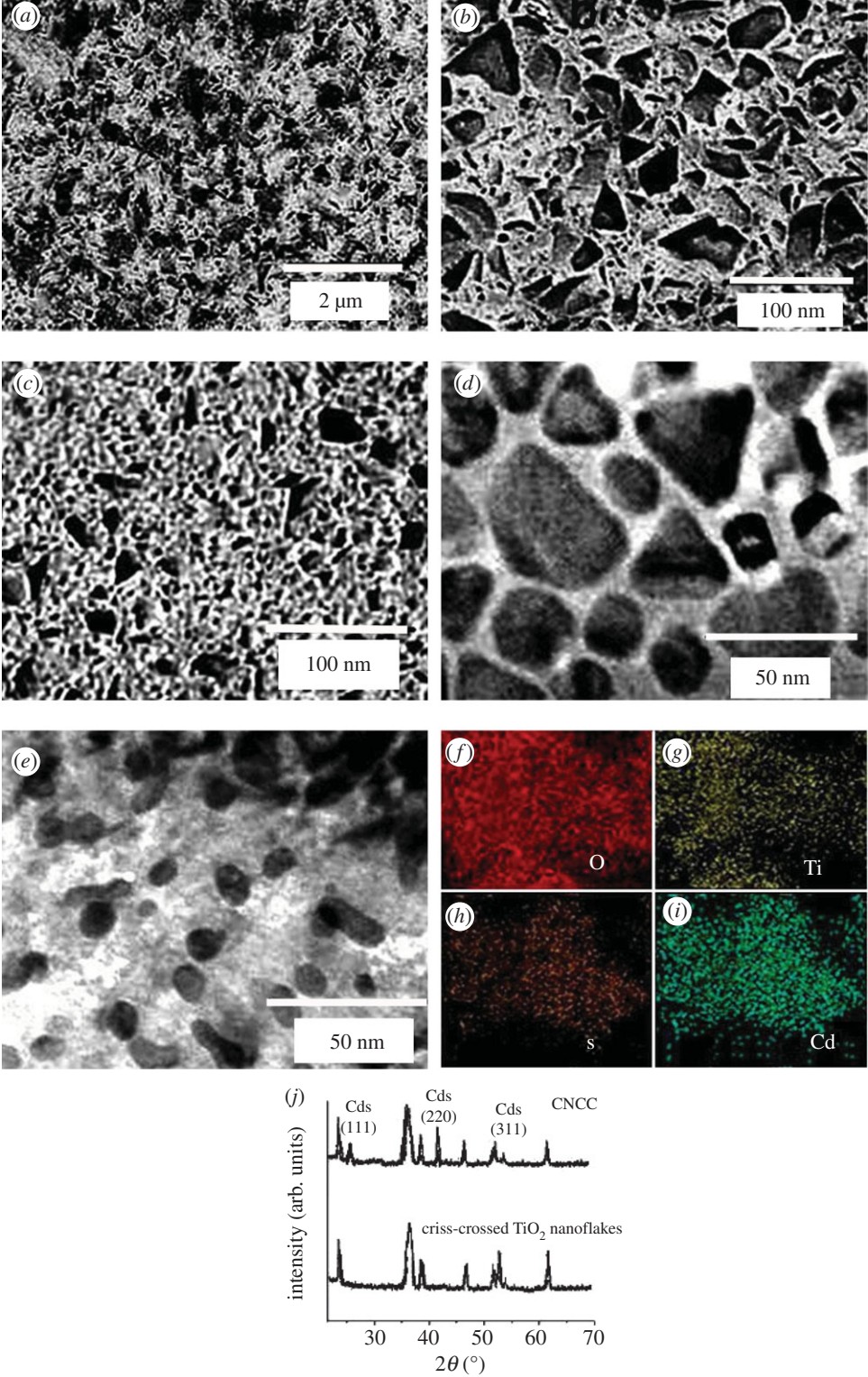

**Figure 1.** Morphology of the CNCC. (*a*) SEM image of the top view of the criss-crossed TiO$_2$ nanoflakes. (*b*) A magnified SEM image of the criss-crossed TiO$_2$ nanoflakes (*c*) A magnified FESEM image of the CNCC. TEM images of TiO$_2$ (*d*) and TiO$_2$–CdS (*e*) composite. (*f*–*i*) Composed elemental mapping image. (*j*) XRD pattern of the CNCC and the criss-crossed TiO$_2$ nanoflakes.

criss-crossed TiO$_2$ nanoflakes are a fine electrical conductive body along the orientation of the strip axes relative to TiO$_2$ nanoparticles. Figure 2*b* shows the recombination time constant of NNCC is two or three orders of magnitude larger than that of NCC in the studied range of light intensity. Slower carrier recombination indicates the criss-crossed TiO$_2$ nanoflakes have less surface recombination sites than TiO$_2$ nanoparticles.

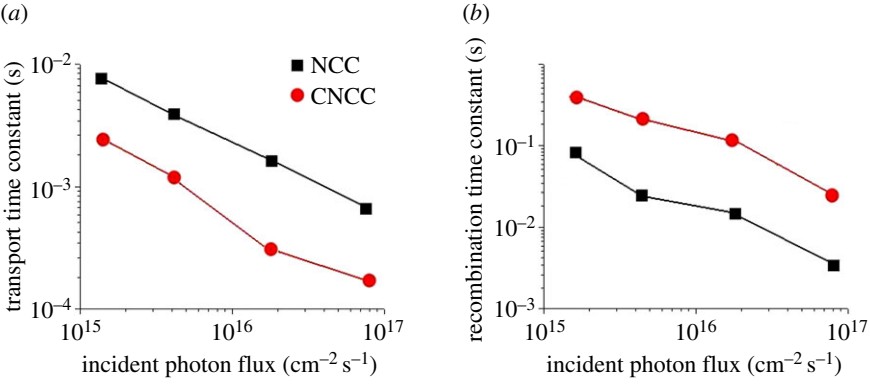

**Figure 2.** Comparison of transmission (a) and recombination (b) time constants for CNCC and NCC.

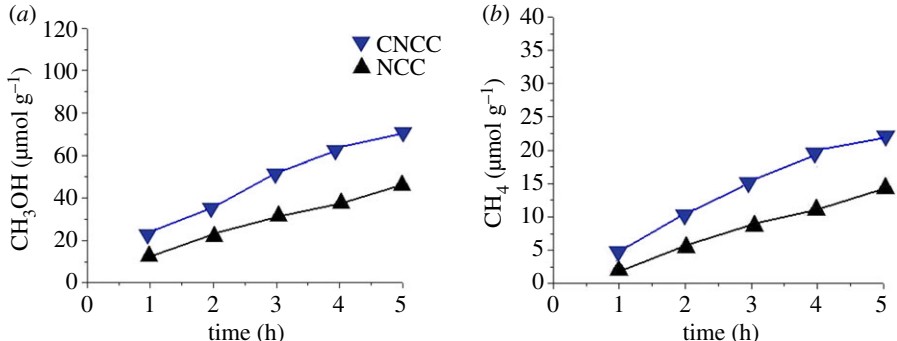

**Figure 3.** Yield of (a) $CH_3OH$ and (b) $CH_4$ upon photoreduction in $CO_2$ as a function of time under visible light by using CNCC and NCC.

## 3.3. Photocatalytic reduction activity

A self-made 100 ml quartz reactor was used for photocatalytic $CO_2$ reduction. A 400 W xenon lamp with a 420 nm cut-off filter was used as the optical source. Gas products from photocatalytic reduction of $CO_2$ were collected with a 1 µm syringe and analysed rapidly by the gas chromatograph (SP-7890) with a flame ionization detector. Detailed experimental procedures of photocatalytic $CO_2$ reduction can be found in supplementary materials. Figure 3 presents the methanol and methane formation rates of NCC and CNCC in 5 h of irradiation. For CNCC and NCC, it is noteworthy that the yield of methanol (figure 3a) is much higher than that of methane (figure 3b). It can be seen that CNCC achieves a methanol formation rate of 24.7 $\mu mol\,g^{-1}\,h^{-1}$, 1.42 times higher than that of NCC. The results show that the methane production rate of CNCC was 4.59 $\mu mol\,g^{-1}\,h^{-1}$, 1.68 times higher than that of NCC. The geometry structure of criss-crossed $TiO_2$ nanoflakes can significantly improve its $CO_2$ photoreduction performance. When visible light irradiates the photocatalyst, CdS acts as a sensitizer at this time, converting the $TiO_2$ response from ultraviolet to visible light. Photoinduced electrons from CdS to the conductive bands of $TiO_2$ can reduce carbon dioxide to negative electrodes, resulting in the formation of methane or methanol. The electron transmission between criss-crossed $TiO_2$ nanoflakes and CdS is faster than that of NCC film, thereby inhibiting recombination with holes.

## 4. Conclusion

The criss-crossed $TiO_2$ nanoflakes combined with CdS were successfully synthesized by a simple solvothermal method using hexadecanol as hydrophobic modifier. The CNCC composites exhibited good $CO_2$ reduction photocatalytic activity under visible light. The results showed that the methane production rate of CNCC was 4.59 $\mu mol\,g^{-1}\,h^{-1}$, which is 1.68 times higher than that of NCC. CNCC achieves a methanol formation rate of 24.7 $\mu mol\,g^{-1}\,h^{-1}$, which is 1.42 times higher than that of NCC. The increase in $CH_4$ and $CH_3OH$ yields can be attributed to the fast electron–hole transmission of $TiO_2$ nanoflakes. This provides a new photocatalytic method for the efficient conversion of $CO_2$ by solar energy.

Data accessibility. Complete data for photocatalytic $CO_2$ reduction are available from the Dryad Digital Repository: https://doi.org/10.5061/dryad.r2f59f0 [18].

Authors' contributions. All three authors participated in all procedures including the design of the study, carrying out the laboratory work, data analysis and drafting the manuscript. All authors approved the final version of the manuscript.

Competing interests. The authors have no competing interests.

Funding. This work was financially supported by Provincial Natural Science, Research Foundation of Anhui Universities, China (KJ2016A880), National Natural Science Foundation of China (No. 21401061), Anhui Provincial Innovation Team of Design and Application of Advanced Energetic Materials (KJ2015TD003).

Acknowledgements. We are grateful to Pinhua Li, who provided suggestions during the drafting of manuscript.

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
