## [Reviewer comments · Royal Society Open Science]

Review History

RSOS-181789.R0 (Original submission)

Review form: Reviewer 1

Is the manuscript scientifically sound in its present form?

No

Are the interpretations and conclusions justified by the results?

No

Is the language acceptable?

No

Is it clear how to access all supporting data?

Not Applicable

Do you have any ethical concerns with this paper?

No

Have you any concerns about statistical analyses in this paper?

No

Recommendation?

Major revision is needed (please make suggestions in comments)

Comments to the Author(s)

The paper is interesting. But author is provided the materials information perfectly and introduction the articles is not significant to highlight their work. So, the present form of the paper is not publishable. After addressing below point successfully, the paper can be considered for publication.

1. Introduction of the paper is not sufficient to highlight the novelty of the work. As for example, TiO₂-CdS composite photo reduction works by other groups are completely hidden. SO, author should describe this part in introduction to show the novelty of their work.
2. Need high resolution FESEM image of Figure 1c. Figure 1b magnification and Figure 1c magnification is not same. It should be the same magnification to compare the morphology of TiO₂ and TiO₂-CdS. CdS particles should be visualized from FESEM image.
3. Need EDX mapping of TiO₂-CdS composite.
4. Need TEM images of TiO₂ and TiO₂-CdS composite to show the CdS real morphology and interface.

Review form: Reviewer 2

Is the manuscript scientifically sound in its present form?

Yes

Are the interpretations and conclusions justified by the results?

Yes

Is the language acceptable?

Yes

Is it clear how to access all supporting data?

Not Applicable

Do you have any ethical concerns with this paper?

No

Have you any concerns about statistical analyses in this paper?

No

Recommendation?

Major revision is needed (please make suggestions in comments)

Comments to the Author(s)

This submission investigated the preparation and characterization of criss-crossed TiO₂ nanoflakes combined with CdS (CNCC) for photo-reduction of CO₂. The topic is interesting and meanful, and the materials and methods were detailed, and the characterization of the CNCC were well performed. Overall, most parts of the manuscript were presented satisfactorily. The modified CNCC and its preparation methods are novel, and the data are reliable. I suggest this

manuscript be accepted for publication in Royal Society Open Science after revisions, and the detailed comments are as follows.

On Abstract

1. Line 16 in Page 1: The authors should re-write this sentence.

On Introduction

2. Line 29 in Page 1: The authors should present the prospective audiences a better background information on photocatalysis, the following two recent articles could serve this purpose in some aspects: Microstructure and performance of Z-scheme photocatalyst of silver phosphate modified by MWCNTs and Cr-doped SrTiO₃ for malachite green degradation, *Applied Catalysis B: Environmental*, 2018, 227: 557-570; Preparation of size-controlled silver phosphate catalysts and their enhanced photocatalysis performance via synergetic effect with MWCNTs and PANI, *Applied Catalysis B: Environmental*, 2019, 245: 71-86.

On Results and Discussion

3. Line 24 in Page 2: On morphological characteristics and phase structures, the authors should consider to discuss more in detail and compare data in this submission with those in other literature including those mentioned above, for example, the first one mentioned above, and the following ones: Insights into atrazine degradation by persulfate activation using composite of nanoscale zero-valent iron and graphene: performances and mechanisms, *Chemical Engineering Journal*, 2018, 341: 126-136; Performances and mechanisms of efficient degradation of atrazine using peroxymonosulfate and ferrate as oxidants, *Chemical Engineering Journal*, 2018, 353: 533-541.

4. Line 24 in Page 2: More other characteristics should be presented to support results of this submission.

5. Line 41 in Page 2: It will be more interesting if the data for the effects of pH and coexistence of inorganic salts were compared with those in the first article mentioned above.

6. On Figure 3 in Page 9: The color of the legend should be designed for the prospective audiences to better read and understand.

7. Line 41 in Page 2: On transportation time and recombination time constants, the authors should consider to cite literatures to support this conclusion.

8. On Reusability: Data on Reusability should be provided.

9. Line 56 in Page 3: On Conclusions, data on CO₂ reduction should be described.

Decision letter (RSOS-181789.R0)

03-Jan-2019

Dear Dr Wang:

Title: Enhanced photoreduction CO₂ efficiency by criss-crossed TiO₂ nanoflakes combined with CdS under visible light

Manuscript ID: RSOS-181789

The editor assigned to your manuscript has now received comments from reviewers. We would like you to revise your paper in accordance with the referee and Subject Editor suggestions which can be found below (not including confidential reports to the Editor). Please note this decision does not guarantee eventual acceptance. I apologise for the delay.

Please submit your revised paper before 26-Jan-2019. Please note that the revision deadline will expire at 00.00am on this date. If we do not hear from you within this time then it will be assumed that the paper has been withdrawn. In exceptional circumstances, extensions may be possible if agreed with the Editorial Office in advance. We do not allow multiple rounds of revision so we urge you to make every effort to fully address all of the comments at this stage. If deemed necessary by the Editors, your manuscript will be sent back to one or more of the original reviewers for assessment. If the original reviewers are not available we may invite new reviewers.

Please also include the following statements alongside the other end statements. As we cannot publish your manuscript without these end statements included, if you feel that a given heading is not relevant to your paper, please nevertheless include the heading and explicitly state that it is not relevant to your work.

- Ethics statement

Please clarify whether you received ethical approval from a local ethics committee to carry out your study. If so please include details of this, including the name of the committee that gave consent in a Research Ethics section after your main text. Please also clarify whether you received informed consent for the participants to participate in the study and state this in your Research Ethics section.

OR

Please clarify whether you obtained the necessary licences and approvals from your institutional animal ethics committee before conducting your research. Please provide details of these licences and approvals in an Animal Ethics section after your main text.

OR

Please clarify whether you obtained the appropriate permissions and licences to conduct the fieldwork detailed in your study. Please provide details of these in your methods section.

Yours sincerely,
Dr Laura Smith

Publishing Editor, Journals

On behalf of the Subject Editor Professor Anthony Stace and the Associate Editor Professor Eva Hevia.

RSC Associate Editor:
Comments to the Author:
(There are no comments.)

RSC Subject Editor:
Comments to the Author:
(There are no comments.)

Reviewers' Comments to Author:
Reviewer: 1

Comments to the Author(s)

The paper is interesting. But author is provided the materials information perfectly and introduction the articles is not significant to highlight their work. So, the present form of the paper is not publishable. After addressing below point successfully, the paper can be considered for publication.

1. Introduction of the paper is not sufficient to highlight the novelty of the work. As for example, TiO₂-CdS composite photo reduction works by other groups are completely hidden. SO, author should describe this part in introduction to show the novelty of their work.
2. Need high resolution FESEM image of Figure 1c. Figure 1b magnification and Figure 1c magnification is not same. It should be the same magnification to compare the morphology of TiO₂ and TiO₂-CdS. CdS particles should be visualized from FESEM image.
3. Need EDX mapping of TiO₂-CdS composite.
4. Need TEM images of TiO₂ and TiO₂-CdS composite to show the CdS real morphology and interface.

Reviewer: 2

Comments to the Author(s)

This submission investigated the preparation and characterization of criss-crossed TiO₂ nanoflakes combined with CdS (CNCC) for photo-reduction of CO₂. The topic is interesting and meaningful, and the materials and methods were detailed, and the characterization of the CNCC were well performed. Overall, most parts of the manuscript were presented satisfactorily. The modified CNCC and its preparation methods are novel, and the data are reliable. I suggest this manuscript be accepted for publication in Royal Society Open Science after revisions, and the detailed comments are as follows.

On Abstract

1. Line 16 in Page 1: The authors should re-write this sentence.

On Introduction

2. Line 29 in Page 1: The authors should present the prospective audiences a better background information on photocatalysis, the following two recent articles could serve this purpose in some aspects: Microstructure and performance of Z-scheme photocatalyst of silver phosphate modified by MWCNTs and Cr-doped SrTiO₃ for malachite green degradation, *Applied Catalysis B: Environmental*, 2018, 227: 557-570; Preparation of size-controlled silver phosphate catalysts and their enhanced photocatalysis performance via synergetic effect with MWCNTs and PANI, *Applied Catalysis B: Environmental*, 2019, 245: 71-86.

On Results and Discussion

3. Line 24 in Page 2: On morphological characteristics and phase structures, the authors should consider to discuss more in detail and compare data in this submission with those in other literature including those mentioned above, for example, the first one mentioned above, and the following ones: Insights into atrazine degradation by persulfate activation using composite of nanoscale zero-valent iron and graphene: performances and mechanisms, *Chemical Engineering Journal*, 2018, 341: 126-136; Performances and mechanisms of efficient degradation of atrazine using peroxymonosulfate and ferrate as oxidants, *Chemical Engineering Journal*, 2018, 353: 533-541.
4. Line 24 in Page 2: More other characteristics should be presented to support results of this submission.
5. Line 41 in Page 2: It will be more interesting if the data for the effects of pH and coexistence of inorganic salts were compared with those in the first article mentioned above.
6. On Figure 3 in Page 9: The color of the legend should be designed for the prospective audiences to better read and understand.
7. Line 41 in Page 2: On transportation time and recombination time constants, the authors should consider to cite literatures to support this conclusion.
8. On Reusability: Data on Reusability should be provided.
9. Line 56 in Page 3: On Conclusions, data on CO₂ reduction should be described.

Author's Response to Decision Letter for (RSOS-181789.R0)

See Appendix A.

RSOS-181789.R1 (Revision)

Review form: Reviewer 1

Is the manuscript scientifically sound in its present form?

Yes

Are the interpretations and conclusions justified by the results?

Yes

Is the language acceptable?

Yes

Is it clear how to access all supporting data?

Not Applicable

Do you have any ethical concerns with this paper?

No

Have you any concerns about statistical analyses in this paper?

No

Recommendation?

Accept as is

Comments to the Author(s)

The author replied all of the raised issue with necessary data. So, paper can be acceptable.

Review form: Reviewer 2

Is the manuscript scientifically sound in its present form?

Yes

Are the interpretations and conclusions justified by the results?

Yes

Is the language acceptable?

Yes

Is it clear how to access all supporting data?

Yes

Do you have any ethical concerns with this paper?

No

Have you any concerns about statistical analyses in this paper?

No

Recommendation?

Accept as is

Comments to the Author(s)

This revised version of the submission could be accepted for publication in Royal Society Open Science now.

Decision letter (RSOS-181789.R1)

15-Feb-2019

Dear Dr Wang:

Title: Enhanced photoreduction CO₂ efficiency by criss-crossed TiO₂ nanoflakes combined with CdS under visible light

Manuscript ID: RSOS-181789.R1

It is a pleasure to accept your manuscript in its current form for publication in Royal Society Open Science. The chemistry content of Royal Society Open Science is published in collaboration with the Royal Society of Chemistry.

On behalf of the Subject Editor Professor Anthony Stace and the Associate Editor Professor Eva Hevia.

RSC Associate Editor:
Comments to the Author:
(There are no comments.)

RSC Subject Editor:
Comments to the Author:
(There are no comments.)

Reviewer(s)' Comments to Author:

Reviewer: 1

Comments to the Author(s)

The author replied all of the raised issue with necessary data. So, paper can be acceptable.

Reviewer: 2

Comments to the Author(s)

This revised version of the submission could be accepted for publication in Royal Society Open Science now.

Appendix A

Response to reviewers

We sincerely thank the reviewers for handling our manuscript and for giving constructive comments. The manuscript has been thoroughly revised according to the comments. The changes made in the revised manuscript are highlighted in **red colour**.

Reviewers' Comments to Author:

Reviewer: 1

The paper is interesting. But author is provided the materials information perfectly and introduction the articles is not significant to highlight their work. So, the present form of the paper is not publishable. After addressing below point successfully, the paper can be considered for publication.

1. Introduction of the paper is not sufficient to highlight the novelty of the work. As for example, TiO₂-CdS composite photo reduction works by other groups are completely hidden. SO, author should describe this part in introduction to show the novelty of their work.

Response: Thanks very much for your suggestion. The introduction part has been re-organized to show the novelty of our work.. TiO₂-CdS composite photo reduction works by other groups have been cited.

2. Need high resolution FESEM image of Figure 1c. Figure 1b magnification and Figure 1c magnification is not same. It should

be the same magnification to compare the morphology of TiO₂ and TiO₂-CdS. CdS particles should be visualized from FESEM image.

Response: We have replaced Figure 1c with high resolution FESEM image. Figure 1b magnification and Figure 1c magnification have changed same. CdS particles have been visualized from FESEM image.

3. Need EDX mapping of TiO₂-CdS composite.

Response: We have added EDX mapping of TiO₂-CdS composite.

4. Need TEM images of TiO₂ and TiO₂-CdS composite to show the CdS real morphology and interface.

Response: We have added TEM images of TiO₂ and TiO₂-CdS composite.

Reviewer: 2

This submission investigated the preparation and characterization of criss-crossed TiO₂ nanoflakes combined with CdS (CNCC) for photo-reduction of CO₂. The topic is interesting and meaningful, and the materials and methods were detailed, and the characterization of the CNCC were well performed. Overall, most parts of the manuscript were presented satisfactorily. The modified CNCC and its preparation methods are novel, and the data are reliable. I suggest this manuscript be accepted for publication in Royal Society Open Science after revisions, and the detailed comments are as follows.

On Abstract

1.Line 16 in Page 1: The authors should re-write this sentence.

Response: We have re-write this sentence.

On Introduction

2.Line 29 in Page 1: The authors should present the prospective audiences a better background information on photocatalysis, the following two recent articles could serve this purpose in some aspects: Microstructure and performance of Z-scheme photocatalyst of silver phosphate modified by MWCNTs and Cr-doped SrTiO₃ for malachite green degradation, Applied Catalysis B: Environmental, 2018, 227: 557-570; Preparation of size-controlled silver phosphate catalysts and their enhanced photocatalysis performance via synergetic effect with MWCNTs and PANI, Applied Catalysis B: Environmental, 2019, 245: 71-86.

Response: Thanks very much for your suggestion. The introduction part has been re-organized.

On Results and Discussion

3.Line 24 in Page 2: On morphological characteristics and phase structures, the authors should consider to discuss more in detail and compare data in this submission with those in other literature including those mentioned above, for example, the first one mentioned above, and the following ones: Insights into atrazine degradation by persulfate activation using composite of nanoscale zero-valent iron and graphene: performances and mechanisms, Chemical Engineering Journal, 2018, 341: 126-136; Performances and mechanisms of efficient degradation of atrazine using peroxymonosulfate and ferrate as oxidants, Chemical Engineering Journal, 2018, 353: 533-541.

Response: Thanks very much for your suggestion. On morphological characteristics and phase structures we have discussed in detail.

4.Line 24 in Page 2: More other characteristics should be presented to support results of this submission.

Response: We have added TEM images of TiO₂ and TiO₂-CdS composite, EDX mapping of TiO₂-CdS composite.

5.Line 41 in Page 2: It will be more interesting if the data for the effects of pH and coexistence of inorganic salts were compared with those in the first article mentioned above.

Response: The types of light sources, wavelength of light sources, wattage of light sources, concentration of solution and gas flow to the reactor used by researchers in photocatalytic reaction are different. Even the photocatalyst loads precious metals has a great influence on the catalytic effect. So this comparison is difficult. However, some researchers compared synthetic photocatalysts with commercial nano-titanium dioxide particles in the same test system. So we compared the recombination time of criss-crossed TiO₂ nanoflakes with that of the typical TiO₂ nanostructured films.

6.On Figure 3 in Page 9: The color of the legend should be designed for the prospective audiences to better read and understand.

Response: On Figure 2 and Figure 3, the colors of the legend have been designed to red and blue.

7.Line 41 in Page 2: On transportation time and recombination time constants, the authors should consider to cite literatures to support this conclusion.

Response: We have cited literatures to support the conclusion.

8.On Reusability: Data on Reusability should be provided.

Response: Data on reusability have been provided.

9.Line 56 in Page 3: On Conclusions, data on CO₂ reduction should be described.

Response: On Conclusions, data on CO₂ reduction have been described.